Tree mortality from a short-duration freezing event and global-change-type drought in a Southwestern piñon-juniper woodland, USA

Poulos Helen M. hpoulos@wesleyan.edu
College of the Environment, Wesleyan University , Middletown, CT , United States
Huston Michael
Electronic publication date: 2014 Jun 10
Publication date: 2014
Volume: 2
Electronic Location ID: e404
Received 2013 Oct 23; Accepted 2014 May 7
Copyright: © 2014 Poulos
Copyright year: 2014
Copyright holder: Poulos
License: This is an open access article distributed under the terms of the Creative Commons Attribution License, which permits unrestricted use, distribution, and reproduction in any medium, provided the original author and source are credited.
License URL: https://creativecommons.org/licenses/by/3.0/

Keywords: Tree mortality, Drought, Piñon-juniper woodlands, Freeze-thaw cycles, Global change, Big Bend National Park

Funding: National Park Service Task Order 4 4-14-08 Funding for this research was provided by the National Park Service under Task Order (J713005001K) 4 4-14-08. The funders had no role in study design, data collection and analysis, decision to publish, or preparation of the manuscript.

==============================
This study documents tree mortality in Big Bend National Park in Texas in response to the most acute one-year drought on record, which occurred following a five-day winter freeze. I estimated changes in forest stand structure and species composition due to freezing and drought in the Chisos Mountains of Big Bend National Park using permanent monitoring plot data. The drought killed over half (63%) of the sampled trees over the entire elevation gradient. Significant mortality occurred in trees up to 20 cm diameter (P < 0.05). Pinus cembroides Zucc. experienced the highest seedling and tree mortality (P < 0.0001) (55% of piñon pines died), and over five times as many standing dead pines were observed in 2012 than in 2009. Juniperus deppeana vonSteudal and Quercus emoryi Leibmann also experienced significant declines in tree density (P < 0.02) (30.9% and 20.7%, respectively). Subsequent droughts under climate change will likely cause even greater damage to trees that survived this record drought, especially if such events follow freezes. The results from this study highlight the vulnerability of trees in the Southwest to climatic change and that future shifts in forest structure can have large-scale community consequences.

Introduction

Recent widespread tree mortality has been documented across the globe in response to increasingly warmer and drier climatic conditions (Allen & Breshears, 1998; Breshears et al., 2009; van Mantgem et al., 2009; Allen et al., 2010). Global-change-type droughts, which are severe droughts coupled with elevated summer temperatures, have resulted in landscape- and regional-scale shifts in forest stand structure and species composition (Breshears et al., 2005; Shaw, Steed & DeBlander, 2005). While multi-year droughts have been widely identified as agents of tree mortality (Guarin & Taylor, 2005; van Mantgem et al., 2009; Ganey & Vojta, 2011), short-duration acute droughts of one to two years in duration can also be responsible for extensive tree death (Breshears et al., 2005; Hogg, Brandt & Michaellian, 2008).

Acute drought events that follow short-duration winter freezes can be especially damaging to plant tissue. Tree death can occur under severe drought after just a single, short-duration freezing event (Willson & Jackson, 2006). Rapid changes in temperature present a unique challenge to trees because cold snaps can cause air bubbles and sap ice to form which can result in stem breakage and hinder water transport (Scholander, Hemmingsen & Garey, 1961; Hammel, 1967; Sucoff, 1969; Zimmermann, 1983).

A five-day freeze occurred in February 2011 in Big Bend National Park, which was followed by the most severe one-year drought on record in Texas in the spring and summer of 2011 (Nielsen-Gammon, 2011) (Fig. 1). West Texas was particularly affected by the drought (National Drought Mitigation Center, 2011), and the Chisos Basin of Big Bend National Park received just 10.9 cm of precipitation in 2011 (one fifth its historical average of 49.2 cm) (WRCC, 2013). Together, the freeze and drought events were likely responsible for widespread tree mortality between 2011 and 2012 in this region.

Figure 1 Climate.

Climatic conditions from 2010 to 2012 in the Chisos Basin of Big Bend National Park, Texas (WRCC, 2013) including (A) monthly extreme low temperatures, (B) mean monthly maximum temperatures, and (C) mean monthly precipitation. The weather station is located within 0.25 km of the middle elevation sample sites in this study.

As part of a permanent forest monitoring study in the Chisos Mountains (CM) of Big Bend National Park, I monitored tree mortality in a Southwestern piñon-juniper forest between 2009 and 2012. This interval overlapped the five-day February freezing event and global-change-type acute drought that occurred in 2011, providing the unique opportunity to document a coupled freezing- and drought-induced tree mortality event. While piñon-juniper tree mortality in response to severe drought has been documented in several sites in the southwestern United States, few studies have examined the combined effects of short-duration freezing and acute drought events on piñon-juniper woodland stand structure and species composition. Moreover, this research highlights tree mortality patterns across a post-Pleistocene relictual mountain range (i.e., Sky Island) that differs dramatically from other previously studied piñon-juniper forests in terms of species composition and climatic setting.

In this paper, I quantify tree mortality by estimating changes in forest stand structure and species composition across the forested area of the elevation gradient in Big Bend National Park. I measured changes in live and standing dead tree density, basal area, and species composition in CM as a whole and at low, middle and high elevations individually. This information provides an assessment of the combined effects of freezing and acute drought stress in Sky Island forests that are surrounded by lowland desert and whose distributions are already greatly restricted by contemporary climatic conditions.

Materials and methods

Study area

The Chisos Mountains are a small rhyolitic mountain range located entirely within Big Bend National Park. Current forests are Pleistocene relicts, and their distributions are the product of species migrations from lowlands to uplands during early Holocene warming (Vandevender & Spaulding, 1979). The CM rise to 2300 m asl. They are bound at lower elevations by deserts dominated by shrub and succulent desert flora, where tree establishment and growth are inhibited due to high temperatures and moisture-limited conditions. The CM represent an ecological transition zone because of their position at the eastern edge of the Basin and Range Province and they share biological affinities with flora of the Rocky Mountains and the Sierra Madre Ranges (Muldavin, 2002). Soils are a mixture of mollisols and entisols. They are composed of moderately deep gravelly loam, which is well drained and non-calcareous (Carter, 1928). Runoff is moderate to rapid. Available water capacity is low.

Forests (above 1600 m asl) in CM are composed of piñon-juniper-oak, pine-oak, and mixed conifer woodlands. Piñon-juniper woodland is the dominant forest type which is comprised of Mexican piñon pine (Pinus cembroides Zuccarini), alligator juniper (Juniperus deppeana vonSteudal), gray oak (Quercus grisea Liebmann), Graves oak (Quercus gravesii Sudworth), Emory oak (Q. emoryi Leibmann), and weeping juniper (J. flaccida vonSchlechtendal) (Poulos & Camp, 2010). Lower elevations also contain small populations of one seed juniper (J. monosperma Englemann) and red berry juniper (J. pinchotii Sudworth) and oak shrublands that are dominated by Q. pungens Leibmann. Arizona pine (P. arizonica Englemann), Douglas fir (Pseudotsuga menziesii Mirbel), and Arizona cypress (Cupressus arizonica Greene) also have restricted populations in Boot Canyon in CM. Taxonomy follows Powell (1998).

The modern climate is arid, characterized by cool winters and warm summers. Precipitation is distributed bi-modally in late summer and winter with the majority of precipitation falling during summer storms as part of the North American Monsoon System. Mean annual precipitation for the Chisos Basin is 49.7 cm (range 10–135 cm). Mean January precipitation is 1.5 cm (range 0–2.5 cm) and is 8.0 cm (range 0.2–20.5 cm) in July. Mean monthly minimum temperatures are 1.8 °C in January and 17.0 °C in July. Maximum temperatures are 14.1 °C in January and 29.1 °C in July.

Field sampling

Thirty-six plots were established at low, middle, and high elevations (12 at each elevation) in the CM in June 2009 and I resampled them during the growing season in June 2012 after the drought. Low elevation plots were randomly placed in Green Gulch within 100 m of the edge of tree cover in CM. Middle elevation plots were randomly distributed across the Chisos Basin. High elevation plots were randomly distributed along the Southeast Rim. Plots were located so that they did not intersect trails, power lines, or archeological or cultural resources. The Southeast Rim was chosen for the high elevation sampling area because it had not previously burned in prescribed fires or wildfires. Trees > 5 cm diameter at breast height (dbh) were measured using 10 m radius (0.03 ha) fixed area plots. Seedlings (individuals < 5 cm dbh) were tallied by species in nested 5 m radius plots. Plot boundaries for both the tree and seedling plots were determined using a two-way ultrasonic rangefinder (Cptcam Inc., Shenzhen, China). The center point of each plot was marked with rebar and its location was recorded with a gps. Each tree was tagged with a uniquely numbered brass tree tag in 2009. I recorded the species, dbh, condition (live or standing dead), distance from the plot center and azimuth from north of each individual. Distance and azimuth measurements greatly assisted in relocating plot center. In 2012, plots were revisited and all trees from the 2009 inventory were resampled. Tree condition (live, recent snag, snag broken above dbh, snag broken below dbh, or clean snag) was noted. Trees lacking leaves or needles, with brittle and/or missing branches were classified as recent snags in the 2012 sampling interval. All recent snags were also checked for evidence of bark beetle infestation including presence of pitch tubes and beetle galleries.

Statistical analysis

I quantified differences in forest stand structure in 2009 and 2012 using linear mixed effects models to account for the repeated measures sampling design. I used the R Statistical Language (R Development Core Team, 2013) and the lme4 (Bates, Martin & Bin, 2012) and lmerTest (Kuznetsova, Christensen & Brockhoff, 2012) packages to perform linear mixed effects analyses of the temporal shifts in forest structure and species composition from the freeze and drought events. Timestep was designated as a fixed effect. Random effects were considered for the intercept, the sample plot, and the interaction of sample plot and timestep. The residuals of each model were inspected for deviations from homoscedasticity, and only models containing residuals without obvious deviations from normality were kept in the analysis. The final structure of the fixed-effects for each model was selected by sequentially dropping non-significant terms from the full model, by measuring changes in the significance of conditional F-tests for each term (Pinheiro & Bates, 2000). The intra-class correlation was also estimated for each model in order to assess the amount of variance in the response variable that can be attributed to the random effects in a model. The models describing the data most adequately were then selected using the Akaike Information Criterion (AIC) (Akaike, 1974). The significance of individual sites and site-year combinations was assessed after final model selection via the F statistic using the lmerTest package.

I used plots as the repeated sampling unit and the sampling year as the treatment representing pre- and post-drought sampling intervals. I compared tree basal area, live seedling and tree density by species, and differences in forest size structure for the two sampling years. I also used mixed effects models to investigate how the drought affected tree populations across the elevation gradient by evaluating changes in tree density and species composition in response to the drought. I evaluated the trend in tree mortality by size by performing a regression analysis comparing the percentage mortality at 1.0 cm size-class intervals.

Results

The 2011 freeze and drought killed over half (62.9%) of the trees in the sample plots in CM. The event triggered significant mortality of both seedlings and trees up to 20 cm dbh (P < 0.05) (Fig. 2). Live tree densities decreased by approximately 100 trees ha−1. Seedlings and smaller trees were preferentially affected by the drought, while larger trees generally survived (Figs. 2C, 2D and 3) (R2 = 0.62; F = 13.1; P = 0.0016). Over half (59.9%) of the seedlings in the monitoring plots died between 2009 (1059 ± 49.8 ha−1) and 2012 (428.8 ± 34.7 ha−1) (P = 0.002). However, basal area also decreased significantly from 12.38 ± 1.75 m2 ha−1 in 2009 to 8.47.6 ± 1.84 m2 ha−1 (P = 0.001) in 2012 indicating that some larger tree mortality also occurred. None of the adult trees that died over the sampling interval showed evidence of bark beetle infestation.

Figure 2 Stand structural change.

Changes in forest stand structure due to drought and freezing in 2011 in the Chisos Mountains, Big Bend National Park, Texas. Mean values (+S.E.) prior to the drought (2009) and after the drought (2011) are shown for (A) seedlings by species, (B) live trees (>5 cm dbh) by species, (C) live trees in 5 cm diameter classes, and (D) standing dead trees. Significant changes between sampling intervals (P < 0.05) are indicated with an (∗). Species codes are as follows: arxa, Arbutus xalapensis; cuar, Cupressus arizonica; frgr, fraxinus greggii; jufl, Juniperus flaccida; jude, Juniperus deppeana; jumo, Juniperus monosperma; pice, Pinus cembroides; prgl, Prosopis glandulosa; quar, Quercus arizonica; quem, Quercus emoryi; qugrav, Quercus gravesii; qugri, Quercus grisea; qupu, Quercus pungens.

Figure 3 Regression.

Regression of tree dbh (cm) as a predictor of percentage tree mortality. Percentage mortality was significantly (P = 0.0016) correlated with tree size (y = 9.9538e−0.062x). Smaller trees suffered 2 to 5 times higher mortality than larger trees.

The freeze and drought resulted in divergent tree mortality patterns among species. Piñon pine experienced the highest seedling and tree mortality (P < 0.0001), and over five times as many standing dead piñon pines were observed in 2012 as in 2009 (54.5% of the piñon pines died). Alligator juniper and Emory oak trees also experienced significant declines in live tree abundance (P < 0.02) (20.7% and 30.9% change in tree density, respectively), and alligator juniper, one seeded juniper, and Emory oak similarly experienced significant seedling mortality (P < 0.05).

Tree mortality occurred across the entire CM elevation gradient (Table 1). Overall tree mortality was significant across all elevations (P < 0.05), and mortality increased with increasing elevation (Fig. 4). Piñon pine experienced significantly greater tree mortality at low elevations (P = 0.007), but otherwise tree mortality by species did not vary significantly over the elevation gradient in response to the freeze and drought.

Figure 4 Elevation mortality.

Changes in mean (+SE) live tree density (ha−1) at low, middle, and high elevations of the Chisos Mountains, Texas. Significant changes between sampling intervals (P < 0.05) are indicated with an (∗).

Table 1 Mortality by elevation.

Changes in live tree density (ha−1) between 2009 and 2011 in the Chisos Mountains of Big Bend National Park, Texas. Values are reported as means (+S.E.).

Elevation	Live trees pre drought	Live trees post-drought	Change in live tree density	
Low	236.6±41.6	146.7±29.1	127.3±23.9	
Midddle	605.1±100.0	483.3±100.0	132.7±60.8	
High	748.4±142.0	502.0±150.0	296.2±82.3	

Discussion

Landscape-scale tree mortality occurred in the Chisos Mountains in response to the five-day February freeze and subsequent global-change-type drought in 2011. The effects of this event spanned the entire mountain range and affected multiple tree species. The tree mortality that occurred in response to this short-duration freezing event and one-year drought is striking because relatively few trees in CM succumbed to the longer decadal drought of the 1990s in this region (H. Poulos, pers. obs., 2012).

While the individual effects of the drought and freezing event could not be distinguished from the present study, both freezing- and drought-induced xylem cavitation likely contributed to the CM tree mortality patterns due to air bubble formation from frozen sap at low temperatures (Pittermann et al., 2005; Sperry, 2011) and to the entry of air bubbles into the xylem conduits across the pit membrane under extremely negative water potentials during the drought (Zimmermann, 1983; Sperry & Tyree, 1990). Pittermann et al. (2005) demonstrated experimentally that conifers exposed to freeze-thaw events occurring in concert with drought stress had high cavitation vulnerability relative to conifers experiencing drought alone. Schaberg et al. (2008) also demonstrated that spring warming following winter freeze caused root damage that resulted in almost 100% seedling mortality in greenhouse experiments on Alaskan yellow cedar. While some have suggested that multiple freeze-thaw cycles are necessary to cause extensive damage to xylem vessels in conifers (Sperry & Sullivan, 1992; Sperry et al., 1994), Willson & Jackson (2006) demonstrated that even conifers with small tracheid diameters like junipers could experience xylem embolism from just a single freeze-thaw cycle when under drought stress. While the drought may have been responsible for most of the tree mortality observed between 2009 and 2012, the visible branch splitting and bark heaving on many CM trees after the freeze (H. Poulos, pers. obs., 2012) indicated that low temperatures during the winter of 2011 could have also contributed to tree death.

Preferential mortality of small trees

With increasing tree size, mortality rate commonly decreases (Lorimer, Dahir & Nordheim, 2001; Palahi et al., 2003). The pattern of higher mortality of smaller trees in CM was consistent with the recent die off event of Pinus edulis between 2002 and 2004 Arizona, New Mexico, Colorado, and Utah, although Mueller et al. (2005) observed the opposite pattern during the 1996 and 2002 acute droughts in piñon-juniper woodlands of northern Arizona. My results in the CM were consistent with the trend of high seedling and sapling mortality under drought relative to larger trees that, with their deeper root systems and larger carbon stores, were able to survive those same drought events (Mendel et al., 1997; Mueller et al., 2005; Lopez & Kursar, 2007; Ganey & Vojta, 2011). The lack of evidence of bark beetle infestation in trees that died over the sampling interval also suggests that the high mortality of small-diameter trees was not related to insect attack.

Differential tree mortality by species

Although Mexican piñon pine is a site generalist in west Texas (Poulos & Berlyn, 2007), the increased mortality of piñon pine relative to other tree species was consistent with the patterns of recent mass tree mortality in the Southwest in 1996 and 2002 where piñon pine was more severely affected by drought than juniper (Mueller et al., 2005; Breshears et al., 2009). Junipers are typically more drought tolerant than pines in the American Southwest (Breshears et al., 2009; McDowell et al., 2008 but see Bowker et al., 2012). So while junipers in CM did experience significant mortality from the 2011 drought, they were probably less affected than the piñon pines because of their higher drought hardiness.

Emory oak was also significantly affected by the drought, and large stands of this species were completely killed in CM. Although southwestern oaks can survive over two months of severe moisture stress under experimental conditions (Poulos & Berlyn, 2007; Ehleringer & Phillips, 1996), little is known about the mechanisms of oak drought and freezing tolerance in the American Southwest (but see Neilson & Wullstein, 1985; Davis, Sperry & Hacke, 1999). Oaks in this region likely display considerable variation in drought and freezing tolerance, but their large tracheid diameters may have led to greater freeze-induced cavitation vulnerability relative to other tree species (Davis, Sperry & Hacke, 1999). Emory oaks experienced lower mortality than piñon pines and junipers in this study, yet, there remains a need for more information about the range of variability in oak drought tolerance mechanisms in the Southwest as they represent a major component of Madrean Sky Island systems.

Shifts in forest stand structure and species composition

Although the mortality event will undoubtedly provide new nesting sites for cavity-nesting birds in CM, the higher mortality of smaller trees, the loss of over half of the piñon pines in my monitoring plots, and the death of piñon pine and entire stands of Emory oak across all elevations could result in major shifts in forest stand structure and species composition. Since 2011, CM has moved out of the drought and is experiencing normal temperature and precipitation levels. The return to normal climatic conditions could have a positive effect on surviving trees by releasing them from competition for moisture and bolstering their survival potential in subsequent droughts (Bowker et al., 2012) since water use efficiency in piñon-juniper woodlands can be associated with stand density (Lajtha & Getz, 1993). Nonetheless, surviving trees in CM may have experienced permanent losses in xylem conductivity in 2011, which could result in delayed tree mortaility (i.e., Bigler et al., 2007) or predispose them to succumb to future acute droughts, especially if these events are coupled with winter freezes. While many piñon pines survived the 2011 drought, future global-change-type droughts could shift CM species towards dominance by junipers and more drought-tolerant oaks.

Mortality patterns across the elevation gradient

The pattern of increased tree mortality with increasing elevation was surprising and contradictory to other prior landscape-scale accounts of drought-induced tree mortality (Allen & Breshears, 1998; Gitlin et al., 2006; McDowell et al., 2008) and canopy dieback (D Schwilk, 2013, unpublished data). The increased tree mortality at higher elevations in CM is probably related to the southerly exposure of the high elevation plots that were located on mesas of the southeast rim at the edge of high elevation forest cover and the exacerbation of the drought effects by the February freeze-thaw cycle. While high elevations of CM are cooler and more humid than low elevations, the South Rim is exposed to high incident solar radiation due to its southerly aspect, as well as high winds and temperature fluctuations because it forms the southern edge of forest cover where the rim drops from 1981 m asl down to the desert floor. Higher elevations also probably experienced the lowest temperatures during the short-duration freeze event in 2011, although cold air drainage also contributes to low temperatures at low elevations (D Schwilk, 2013, unpublished data). This may have stimulated greater damage to high elevation trees through freezing-induced xylem cavitation in high elevation trees which may have let to higher mortality during the course of the drought.

Conclusion

The results from this study demonstrate the impact of freeze-thaw events followed by drought on Sky Island forest stand structure and species composition. Future acute drought events are likely to occur with greater frequency as global mean temperatures rise in the coming decades, and the climate becomes more unpredictable (Jentsch, Kreyling & Beierkuhnlein, 2007). Subsequent droughts are likely to cause even greater damage to trees that survived this record drought in Texas, especially if future drought events are coupled with severe freezes. Although I documented significant rapid tree mortality in CM over the study period, lagged tree mortality is likely. Delayed mortality has been observed elsewhere in response to severe drought (Pedersen, 1998; Bigler et al., 2007), since damage to water transport tissue can occur over multiple years (Tyree & Sperry, 1989; Hanson & Weltzin, 2000) and because tissue damage scan also predispose trees to subsequent mortality from beetle infestations (Allen & Breshears, 1998). The dramatic tree die off in CM in response to just one year of abnormal climatic conditions highlights the need for long-term forest monitoring and studies that predict the effects of future climatic extremes on Sky Island forests of the American Southwest.

The author thanks Richard Gatewood of the National Park Service for logistical assistance. Darren Wallis, Peter Stothart, and Leslie Kuhn provided valuable field assistance for this project.

Additional Information and Declarations

Competing Interests

Author Contributions

The authors declare there are no competing interests.

Helen M. Poulos conceived and designed the experiments, performed the experiments, analyzed the data, contributed reagents/materials/analysis tools, wrote the paper, prepared figures and/or tables, reviewed drafts of the paper, was solely responsible for the content of the manuscript.

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
