# Peer review of "Tree mortality from a short-duration freezing event and global-change-type drought in a Southwestern piñon-juniper woodland, USA"

_PeerJ, doi:10.7717/peerj.404_

## Round 0.1 · original submission · Major Revisions

Dear Helen,

Both reviewers were very positive about the value of your study. However, both have made a number of suggestions that should help you revise it to have the impact it deserves.

I think you will find their comments clear and helpful, and I look forward to seeing a revision soon.

Reviewer 1 ·

Basic reporting

No comments

Experimental design

More details are needed in the description of methods. I give specific comments on this in the author comments.

Validity of the findings

I question some of specific conclusions given in the author comments and suggest ways to fix this. I feel there is a bit too much speculation on tree species physiology of mortality, since this study was not designed to investigate that. More details are in the general comments.

Additional comments

This study documents an important recent event, the Texas drought of 2011, and how it affected woodland tree species in western Texas. The study is timely, and it’s very fortuitous design (plots were installed and measured prior to the drought), has led to some great insight into the potential effect of climate change on the trees of this regions. I have a number of general and specific comments on how to improve the manuscript:

General comments:

1. There are too many words in the manuscript devoted to speculation on the physiological processes of how these trees die from drought. While tree drought mortality of physiology is a subject of great research interest and activity, and some physiological insight on the species studied is relevant, the data in this study do not address that problem. Instead, the author should devote more space to discussing the very useful data that were collected on the patterns of which trees died where. This problem starts in the abstracts and continues throughout the paper; much of section 4.3 is an example of where this change is needed.
2. Species differences in mortality responses are barely covered in the manuscript (despite section 4.3). How did the mortality event shift the tree community composition across elevations? From Fig 1b it is apparent that most trees which died were P. cembroides, yet other species may have had their populations cut in half. Was this mortality also uniform across elevations?
3. What was the role of bark beetles in the piñon mortality? Does Ips confusus attack P. cembroides? Were trees checked for insect damage or signs of bark beetle attack? Interestingly, your mortality distribution by size class data may offer insight on this. Although it may not be known for piñon Ips, mountain pine beetle shows a clear size preference in attacked trees (around ~10-15 cm DBH in Axelson et al 2009, For. Ecol. Mgmt.). Results of Mueller et al. (that you discuss on Line 161) were consistent with a bark beetle preference for larger trees and the beetles are a well-documented component of the P. edulis mortality event in Arizona and the 4 corners region. Your data are not consistent with this pattern, perhaps suggesting a minimal role of bark beetle attack in mortality in the Chisos.
4. The order of the drought and freeze is not clear. On line 45, the author states that the drought was coupled with the freeze. But didn’t the drought of 2011 mostly follow the freeze. In much of Texas, I believe this drought was a reduction in summer rainfall that year, but this may not be the case in the bi-modal southwest of the state. Including a figure showing the Chisos climate (temperature and precipitation) from 2009-2012 (the period of the study) would greatly improve the paper and clear this up. This could be done with the WRCC data or from monthly PRISM data if there are no stations close to the mountains. Be sure to indicate the frost on such a figure, it may not show up if monthly averages are used. More details on the sequence of climate events would also be helpful in the methods section.

Specific comments:

Line 1 and elsewhere: I believe “global-change type drought” should be “global-change-type drought” to be grammatically correct. This is the usage in Breshears et al. 2005.

Line 15: Invoking hydraulic failure here is fairly speculative and not useful. While hydraulic failure is not improbable (nor is carbon starvation, nor any other mortality mechanism), the data presented in this manuscript do not address the physiology of how these trees died from drought. To state this here is to imply that the data on this will be presented to the reader, which is misleading.

Lines 16-17, 176, 184: There is no data in this study, nor in any of the others that demonstrates or suggests that P. cembroides is isohydric. This is assumed by the author since P. edulis, its better-studied piñon pine cousin, has this behavior. The assumption is reasonable, but it’s also reasonable to assume that oaks in this study are also isohydric, since other oaks have been demonstrated this drought response. Therefore the hypothesis the P. cembroides was more severely affected because it is isohydric is not a good one, since the less-affected oaks are also isohydric.

Line 30-31: Also see Bigler et al., 2007, Oikos, who demonstrated multi-year lagged mortality responses to drought events.

Line 41: Regarding the combined effects of drought and freezing on tree mortality see also Schaberg et al., 2008, Global Change Biology (and other papers by this group), who implicate these combined factors in the widespread mortality of yellow-cedar in Alaska.

Line 45: Coupled is not the right word here. Wasn’t the 2011 summer drought preceded by the February 2011 freeze?

Line 59: remove “accelerated”.

Line 63: “Measure” should be “measured”

Line 81: “medium” should be “moderate”

Line 83: Awkward sentence due to use of dominant and dominated.

Line 100: How many plots were established and re-measured? What was their spatial extent and elevational distribution? Include a map if necessary.

Line 102: How were plots selected? Are they arranged along a transect? Chose randomly from a grid? Arbitrarily arranged? Systematically selected by elevation?

Line 121: Very cool result!

Line 132-133: Also a very interesting result, consistent with the P. edulis mortality event of 2002-2004. Consider adding a figure showing mortality rates across elevations. Is there a relationship between size distribution and elevation? If there are more seedlings at higher elevations, could this cause the elevation trend?

Line 136: Was this really measured over a widespread area? See earlier comment on Line 100. Perhaps you mean “severe” or a “high-degree” of tree mortality? Widespread tree mortality certainly affected Texas in 2011.

Line 138: Awkward sentence. “The response to this intense one-year drought and short-duration freezing event is striking because relatively few trees in CM succumbed …” would be an easy fix.

Line 143: This paragraph on xylem cavitation and drought mortality works well due to the disclaimer in the first sentence regarding the limits of the study, and the careful language: “likely contributed”. In addressing general comment #1 above, these sentences could be kept.
Line 153: However, did the freeze occur before the drought started in this study? See general comment #4.

Line 161: used “following” instead of “during” the mortality, and that study, occurred after the drought.

Line 166: The pattern of mortality across size observed by Mueller et al., was also consistent with bark beetle preference – see general comment #3.

Line 171: Instead of “harder hit” use “more greatly affected” or similar.

Line 172: Piñon, oaks, and junipers in the CM all loose hydraulic conductivity every day in the hot, semi-arid west (piñon is at 25% PLC by -2 MPa, following Linton et al. 1998). Do you mean a catastrophic and/or unrecoverable loss here?

Lines 174-182: The link between isohydric stomatal regulation and physiological mortality mechanism is a hypothesis proposed by McDowell et al., 2008. The data from studies that actually test this hypothesis show the picture is not as clear, P. edulis experienced bigger changes in hydraulic function than in carbohydrates in mortality experiments (Adams et al. 2013, New Phytol.; Sevanto et al. 2013, PCE). There is little published data for this on the anisohydric juniper. Additionally, the ideas on what carbon starvation would even look like have advanced much since McDowell 2008 – see Sala et al. 2010, New Phytol.; Sala et al. 2012, Tree Phys; McDowell et al. 2011, TREE; and many others. You are not up to speed here on this literature and not adding anything new to this discussion. For these reasons I suggest re-writing this section to be less speculative and more focused on the data you did collect – see general comment #1.

Lines 182: In 2009 and 2012 there were more piñon than juniper in the plots, but both populations were reduced by 2012. From figure 1b this looks like a 50% reduction for piñon, and a 25% reduction for juniper. You should state these numbers in any comparison.

Lines 187 and 200: Interesting! – but what does the pattern of Emory oak mortality look like across elevations? What is this pattern for all species? I think a figure comparing this (and its discussion) would be really insightful and would greatly improve the manuscript.

Line 193: Were Emory oaks relatively less affected? What was the % change in the population? From fig 1b it looks like ~50% - similar to piñon pine. Given this statement, and the piñon-juniper comparison above, the paper would benefit from a figure showing relative differences in population reduction – a figure like 1b, but with % change or % reduction in trees/ha on the y-axis.

Line 194: Speculating that Emory oak are anisohydric because they were not affected much is completely nonsensical. Besides the issue raised in the previous comment, hydraulic strategy cannot be simply assumed to relate to drought mortality – as stated earlier in the paper, anisohydric species may be vulnerable. Don’t forget that P. cembroides is assumed to be isohydric -there is no data on this shown or cited. So you are making assumptions about hydraulic strategy differently for piñon (because it’s related to other isohydric piñon) and oak (because of its perceived drought tolerance) and then finding that these are different. Such speculation is not useful and should be removed.

Line 221: Again, didn’t the freeze come before the drought? This is how it’s written on lines 226-228. This needs to be consistent throughout and illustrated in a figure – see general comment #4

Line 228: What were Waring and Schwilk's results? You are implying they came to a different conclusion about drought effects, but it’s not clear what that is. If you want to relate your results to their study, you will need to discuss their study in more detail. I see from your submission comment that Waring and Schwilk’s paper is also in review at PeerJ – but I was not given access to it.

Line 237: See also Figure 3 in Jentsch et al. 2007, Frontiers in Ecol. Env. Cold extremes may not disappear entirely with climate change, as the distribution of climate parameters is expected to widen.

Table 1: Are these numbers in trees/plot or trees/ha? Label this. It must be a mean since there is a calculated SE. Also, the far right column shows reduction in live tree density. Change in live tree density would show negative values.

Figures: Larger fonts are needed in the figures.

·

Basic reporting

The manuscript is well written and the raw data presented in a clear manner, although the statistical analysis is, as far as I can tell, is inadequate. For the purpose of completeness, the authors should also include information on the climate conditions of 2009, which was dry in parts of Texas, though not in West Texas. At any rate, readers cannot be expected to know that.

Experimental design

The description of the experimental design is not sufficiently detailed. How many 10 m radius plots were censuses and what was there spatial layout along the elevation gradient? Where the nested 5 m radius plots marked for resampling or was another nested subsample used both times? How many seedlings and adult trees were counted in total?

The description of the statistical analyses is likewise not sufficiently detailed. How was “stand structure” quantified for the t-tests? Was each plot comparison done as a separate t-test? I would suggest that a collection of t-test is not the best way to examine this data set. Consider analyzing larger collections of data in a Generalized Linear Model using a negative binomial distribution on the count data, and elevation, species and dbh as factors or correlates of mortality.

Validity of the findings

As you report F values in the results, I surmise that you used some kind of ANOVA analysis. You must specify exactly how they were done, for this information to be of use.

Additional comments

This is a very good and interesting data set. With a little more work on the manuscript, it will become a valuable contribution to the growing literature on drought mortality

---

## Round 0.2 · Minor Revisions

PeerJ 2013:10:931:1:2: Helen M. Poulos “Tree mortality from a short-duration freezing event and global-change-type drought in a Southwestern pinon-juniper woodland, USA”

Dear Dr. Poulos,

I have looked over your revised manuscript and concluded that you have satisfactorily addressed the reviewers’ comments and produced a much improved manuscript.

However, in re-reading your manuscript I have noted a number of typos, grammatical errors, and unclear word usages that need to be corrected before publication. The suggested wording is indicated below

line 14 “were observed in 2012 as in 2009”

line 26 “Global-change-type droughts, which are severe droughts coupled with elevated summer temperatures (Breshears et al. 2005),....”

line 31 “can also be responsible for...”

line 65 I am unaware of any Sonoran desert anywhere near the Chisos Mountain. It is misleading to include “Sonoran” in this sentence, which should be expressed as “...that are surrounded by lowland desert and whose....

line 123 ...’mixed effects analyses of the temporal shifts in forest structure...”

line 144 This paper has a single author. Why do your refer to “our sample plots.” If another group or more people are involved, this should be explained in the methods, if they are not listed as authors. It would be simpler and clearer to say “...of the trees in the sample plots....”

line 145 “...triggered significant mortality of both seedlings and trees...”

line 150 please double-check your calculation of basal area. 40 – 65 m2 ha sounds awfully high to me for these arid forests, unless you sampled in dense “dog-hair” stands. Most of the South Rim looks like open savanna, rather than dense forest. Perhaps you should include a map of the locations of your study sites.

line 155 “...were observed in 2012 as in 2009...”

line 162 “Pinon pine experienced significantly greater mortality at low elevations...”

line 172 “... individual effects of the drought and freezing event could not be distinguished in the present study...”

lines 185-190 This entire paragraph reviews published results showing that prior freezing exacerbates the negative effects of drought on trees. So it make no sense to conclude with a sentence that states “This suggests that drought probably had the greatest effect on the mountain range-wide mortality .... “ The information you presented includes nothing to indicate whether the effects of freezing or drought are greater. This sentence should either be deleted or modified to something like this:

“While the drought may have been responsible for most of the tree mortality observed between 2009 and 2012, the visible branch splitting and bark heaving ...... indicated that low temperatures during the winter of 2011 likely also contributed to tree death. It seems unnecessary to mention Schwilk at this point. He is mentioned earlier.

line 217 “...their large tracheid diameters may have led to greater...”

line 231 “Nonetheless, surviving trees in CM may...”

line 236 “... by junipers and more drought-tolerant oak species.”

line 243 “...high elevation forest cover and to the exacerbation of the drought effects...”

lines 250-251 “...xylem cavitation which may have led to higher mortality during the subsequent drought.” DELETE the following sentence “This contradicts Waring and Schwilk’s...of woody life forms.” If there is a contradiction, it would require a much more extensive discussion to resolve and is not appropriate as a single sentence at the end of your discussion.

line 257 “... followed by drought on Sky Island....”

line 261 there has been no mention of “freeze-thaw cycles” in any of the previous text. This should be rewritten as something like “...especially if future droughts are coupled with severe freezes.”

line 265 “... and because tissue damage can also predispose...”


NOTE: I did not check the literature for either format or completeness. However, I did notice that the reference WRCC 2013) in line 403 is not in the literature cited.

Table 1. The third column “change in live tree density” should be expressed as percentage and labeled as “Reduction in live tree density.”

Figure 1. The color used for the lines for 2012 data do not show up when printed in black and white. Another color should be chosen so people who print B&W copies can read the graphs.

line 410 Figure 2 legend. “Significant changes (p < 0.?????) between sampling intervals are indicated with an (*).”

line 411 Figure 3 legend. This is a regression on tree size, NOT on “tree size class.” The legend should be rewritten as “Tree dbh (cm) as a predictor of tree mortality. Percentage mortality was significantly (p < 0.?????) correlated with tree size ([PROVIDE Regression equation at this point in the legend]). Smaller trees suffered... than larger trees.” ALSO, the axis legends of the figures should be changes to: “Percentage Mortality” and “Tree Diameter (cm)”.

line 415 Figure 4. Indicate p value for significant changes between sampling intervals.

PLEASE carefully proofread your manuscript and check for completeness and accuracy of the references before submitting the final version.

---

## Round 0.3 · Minor Revisions

We're almost there. I've read your revision and it looks fine, but there are a few more things to do before final acceptance.

The two remaining major issues should be very easy to take care of.

1) The legend for figure 2 should include the species names associated with the abbreviations you use on the X axis in A and B, as well as the common names. Alternatively you could include this information in a table that might include a bit more information about the species, such elevational range, mean and sd of density, basal area, % mortality, etc. Since you discuss some of these details in the text, you could easily put in a reference to Table 2.

2) The quality of the figures. I've looked at the PNGs you sent, and they are all at too low a resolution for publication, they look fuzzy in my printed hardcopy, and they look fuzzy on the screen. You should either use a PNG, JPEG, or other format at higher resolution (dots per inch, presumably Peerj recommends some minimum standard) or include them as generated lines, as in a pdf.

Minor editing:

line 24 - don't capitalize "the" in "the Chisos Mountains"

line 186 = no comma after "although" lower case "s" in "southwestern"

Thanks for your work on this manuscript. It's very interesting and clearly written.

---

## Round 0.4 · accepted · Accept

Thanks for taking care of these details. I'll be glad to see this out and getting some attention.